# Evaluating Gelatin-Based Films with Graphene Nanoparticles for Wound Healing Applications

**DOI:** 10.3390/nano13233068

**Published:** 2023-12-02

**Authors:** Piotr Kamedulski, Marcin Wekwejt, Lidia Zasada, Anna Ronowska, Anna Michno, Dorota Chmielniak, Paweł Binkowski, Jerzy P. Łukaszewicz, Beata Kaczmarek-Szczepańska

**Affiliations:** 1Department of Materials Chemistry, Adsorption and Catalysis, Faculty of Chemistry, Nicolaus Copernicus University in Torun, Gagarina 7, 87-100 Torun, Poland; pkamedulski@umk.pl (P.K.); 302353@stud.umk.pl (P.B.); jerzy_lukaszewicz@o2.pl (J.P.Ł.); 2Centre for Modern Interdisciplinary Technologies, Nicolaus Copernicus University in Torun, Wilenska 4, 87-100 Torun, Poland; 3Department of Biomaterials Technology, Faculty of Mechanical Engineering and Ship Technology, Gdańsk University of Technology, Gabriela Narutowicza 11/12, 80-229 Gdansk, Poland; marcin.wekwejt@pg.edu.pl; 4Department of Biomaterials and Cosmetics Chemistry, Faculty of Chemistry, Nicolaus Copernicus University in Torun, Gagarina 7, 87-100 Torun, Poland; 503555@doktorant.umk.pl (L.Z.); 317323@stud.umk.pl (D.C.); 5Department of Laboratory Medicine, Medical University of Gdańsk, Marii Skłodowskiej-Curie 3a, 80-210 Gdansk, Poland; anna.ronowska@gumed.edu.pl (A.R.); anna.michno@gumed.edu.pl (A.M.)

**Keywords:** graphene nanoplatelets, gelatin film, porous carbon, wound dressing

## Abstract

In this study, gelatin-based films containing graphene nanoparticles were obtained. Nanoparticles were taken from four chosen commercial graphene nanoplatelets with different surface areas, such as 150 m^2^/g, 300 m^2^/g, 500 m^2^/g, and 750 m^2^/g, obtained in different conditions. Their morphology was observed using SEM with STEM mode; porosity, Raman spectra and elemental analysis were checked; and biological properties, such as hemolysis and cytotoxicity, were evaluated. Then, the selected biocompatible nanoparticles were used as the gelatin film modification with 10% concentration. As a result of solvent evaporation, homogeneous thin films were obtained. The surface’s properties, mechanical strength, antioxidant activity, and water vapor permeation rate were examined to select the appropriate film for biomedical applications. We found that the addition of graphene nanoplatelets had a significant effect on the properties of materials, improving surface roughness, surface free energy, antioxidant activity, tensile strength, and Young’s modulus. For the most favorable candidate for wound dressing applications, we chose a gelatin film containing nanoparticles with a surface area of 500 m^2^/g.

## 1. Introduction

The development of civilization, a sedentary lifestyle, poor nutrition, and the accompanying stress definitely have a negative impact on the quality of life in today’s society. As our health deteriorates, humans become more vulnerable to various diseases and other problems. Wounds—often resulting from mechanical or thermal injuries—are defined as damage or tears to the skin’s surface, and are treated with various dressings [1].

Natural polymers are biodegradable, biocompatible and non-toxic [2,3], which is of great importance to their use in medicine and as a dressing for damaged skin. Biopolymer dressings are necessary to heal skin injuries and reconstruct damaged tissues [4]. Among the various dressings used in the biomedical area, biopolymers have shown significant potential for application in effective wound healing by providing a moist environment at the injury interface and enabling oxygen exchange between fabrics and the external environment [5]. Additionally, this type of dressing is often used as a carrier of active compounds, such as antibacterial agents, anti-inflammatory substances, etc., which additionally support tissue regeneration [6].

Graphene is an emerging material in electronic and energy applications, including acting as an electrode material in batteries, supercapacitors, and fuel cells. Despite the growing interest in graphene research, some specific domains still need to be adequately explored. Bio-oriented (medicine, cosmetics, etc.) applications of graphene belong to such underestimated fields. According to some previous studies [7], 3D-structured graphene flakes exhibited biocompatibility with blood cells (DPPH tests, blood compatibility), which opens the field of potential application in medicine. Graphene in its pristine form is a 2D material, however; this geometrical form is not preferred in some experiments, such as drug delivery or biofiltration. For such purposes, a well-developed pore structure is needed, emphasizing the importance of all measures converting loose graphene flakes into a porous material with a permanent pore structure. Some methods have been established [8,9] which result in the surface area and pore structure development reaching more desirable values (ca. 1000 m^2^/g and above 1 cm^3^/g, respectively). 

Gelatin is a well-known biopolymer of animal origin, obtained by partial hydrolysis of collagen [10]. Further, due to its biocompatibility, biodegradability, and non-toxicity (the material is recognized as safe by the US Food and Drug Administration (US FDA) [11], it is widely used in the various fields of medicine, among others, in wound dressings, tissue engineering, and surgical adhesives. As knowledge has developed, scientists have always looked for techniques that accelerate the tissue regeneration process, which is why adding bioactive nanoparticles to the construction of biopolymers is becoming more and more common. For example, biopolymers with added selected metallic nanoparticles significantly affect the effectiveness of antibacterial agents by inhibiting the possibility of wound infection, significantly improving damaged tissue’s healing process [4]. Furthermore, the use of graphene nanoparticles can be extensive. The potential use of graphene in modern technology is influenced by aspects such as the filtering properties, strength, and flexibility of a material with a two-dimensional structure. Materials based on graphene are nanomaterials that exhibit good biocompatibility and broad-spectrum antibacterial activity that can interact with other biological molecules, such as proteins, enzymes, and other factors [12]. However, there are also some reports about their cytotoxicity, which is dependent mainly on size, concentration, and exposure duration [13]. This mechanism is attributed to the generation of reactive oxygen stress, which can cause DNA damage or disturb cell signaling [14]. For example, Shvedova et al. [15] reported that carbon derivatives may result in skin irritation and diseases after cutaneous exposure. Further, it is generally accepted that graphene shows higher cytotoxicity than graphene oxide, related mainly to its aggregation tendency [16].

The aim of this study was to obtain and characterize novel gelatin-based materials in thin film form, modified with special graphene nanoparticles, for application as wound dressing. The characterization of graphene nanoparticles was carried out. Also, their biocompatibility with human blood and cells was determined. Nanoparticles without the toxic effect were selected and added to gelatin to fabricate thin films.

## 2. Materials and Methods

### 2.1. Materials

Commercial materials for further investigations, i.e., graphene-type powder materials and porcine-derived gelatine, were delivered by Sigma-Aldrich (Poznań, Poland). Unless otherwise noted, reagents for hemo- and cytocompatibility studies came from Merck KGaA (Darmstadt, Germany). For better understanding, some symbolic names for the obtained samples were proposed according to the general formula X-Y. X-Y describes the type of graphene material: low surface area GF-15 (150 m^2^/g), medium surface area GF-30 (300 m^2^/g), and high surface area GF-75 (750 m^2^/g).

### 2.2. Graphene Nanoparticle Characterization

A scanning electron microscope (SEM, 1430 VP, LEO Electron Microscopy Ltd., Oberkochen, Germany), capable of working in STEM mode (detecting BF and DF), was applied to determine the structure of the investigated materials. Surface area and porosity studies were performed by means of a widely approved method of low temperature (−196 °C) adsorption of nitrogen. An automatic sorptometer was used for this purpose, i.e., ASAP 2010 (Micromeritics, Norcross, GA, USA). Each analysis was preceded by high temperature desorption in a vacuum at 200 °C for 12 h. All determined nitrogen adsorption isotherms were considered the II type, according to the IUPAC. In such a case, it is assumed that the nitrogen adsorption follows the monolayered–multilayered mechanism.

Additional instrumental studies were performed to acquire information on the elemental composition of CHN (Vario MACRO CHN, Elementar Analysensysteme GmbH, Langenselbold, Germany), and graphene deglomeration (micro-Raman spectrometer (laser wavelength 532 nm), Senterra, Bruker Optik, Billerica, MA, USA). Crucial Raman anlysis parameters were set as: careful focusing through a 50× microscope objective; excitation power, 2 mW; resolution, 4 cm^−1^; CCD temperature, 223 K; laser beam width, 2.0 µm; and an integration time of 100 s (50 × 2 s).

### 2.3. In Vitro Biocompatibility

The in vitro studies on hemo- and cytocompatibility of nanoparticles were conducted on red blood cells (RBCs) and fetal osteoblast cells (hFOB 1.19, ATCC, Manassas, VA, USA) of human origin. To determine the number of cells, a hemocytometer Superior CE (Marienfeld, Lauda-Königshofen, Germany) was used. Before testing, powders were sterilized through 30 min of UV light exposure.

#### 2.3.1. Hemocompatibility

RBCs were isolated and fractionated according to the standard protocol [14], as a by-product from buffy coats obtained during the blood donation from healthy volunteers at the Regional Centre in Gdańsk (under the approval of the Regional Bank Review Board, with the institutional permission M-073/17/JJ/11). RBCs (3 × 10^9^ cells/mL) were incubated with the nanoparticle powders (n = 3; 100 mg/3 mL) at 37 °C for up to 24 h. Then, the suspensions were centrifuged to obtain supernatants for 3 min at 100× *g* at room temperature. The hemolysis (expressed as a percentage) was measured using an Ultrospect 3000pro spectrophotometer (Amersham-Pharmacia-Biotech, Cambridge, UK) at a 540 nm wavelength. For a positive control, RBCs were treated with 0.2% Triton (i.e., 100% hemolysis), while for a negative control, RBCs were incubated without nanoparticles. According to the literature, materials resulting in hemolysis below 2% are nonhemolytic [17].

#### 2.3.2. Cytocompatibility

For the study, extracts from the tested nanoparticles (n = 3; 100 mg/1.5 mL) were prepared through a direct extraction method, according to ISO 10993-5 [18]. The osteoblast cells (hFOB 1.19) were grown in a culture medium based on Ham’s F12 Medium and Dulbecco’s Modified Eagle’s Medium (without phenol red), in the proportion 1:1, containing L-glutamine (1 mmol/L), geneticin (G418; 0.3 mg/mL), and 10% fetal bovine serum. The cell culture was carried out at 37 °C in a humidified atmosphere with 5% CO_2_. Then, cells at a density of 12 × 10^3^ were seeded on a 96-well plate and incubated until a confluent layer was obtained. Next, the culture medium was exchanged on those containing tested extracts. The viability of hFOB cells was evaluated after 24 h, using the MTT assay (3-(4,5-dimethylthiazol-2-yl)-2,5-diphenyltetrazolium bromide; 0.60 mmol/L) spectrophotometrically at 570 nm wavelength. The results were presented as a % of change, referring to the living cells grown on the tissue culture plate (TCP, 100%). Further, the LDH assay, which determined the death of cells during the culture, was tested by directly measuring the NAD oxidation of lactate dehydrogenase (LDH) spectrophotometrically at a 340 nm wavelength. The results were presented as a % of the total LDH released from the cells grown on TCP. According to the ISO standard, reducing cell viability by more than 30% is considered a cytotoxic effect [18].

### 2.4. Film Preparation and Characterization

Gelatin was dissolved in distilled water at 1 w/w% concentration. Selected graphene nanoparticles were added to the gelatin solution at 10 w/w% concentration, which is the lowest concentration that allows the creation of homogeneous films. The mixture was mixed with a magnetic stirrer for 1 h (400 rpm) and then placed in plastic holders (40 mL per 10 cm × 10 cm) to evaporate the solvent (room conditions, 72 h). Thin films—with 0.017 mm (±0.003) thickness, measured with a gauge (Sylvac, Valbirse, Switzerland)—were obtained. Gelatin film without graphene nanoparticles was studied as a control. Further, the films were denoted as: Gel_X-Y.

#### 2.4.1. Scanning Electron Microscope (SEM)

A scanning electron microscope (SEM; LEO Electron Microscopy Ltd., Cambridge, UK) was used to observe the surface and cross-section morphology of the obtained films.

#### 2.4.2. Mechanical Properties

The mechanical properties were tested using a universal testing machine (Shimadzu EZ-Test EZSX, Kyoto, Japan) in stretching mode (initial force at 0.1 MPa, crosshead speed fixed at 5 mm/min, n = 10) [19]. The samples were cut using a paddle-shaped stencil and a hand press. The mechanical parameters, such as Young’s modulus, maximum tensile strength, and elongation at break, were calculated using the Trapezium X Texture program.

#### 2.4.3. Antioxidant Activity

The antioxidant properties of the films were determined using the 2,2-Diphenyl-1-picrylhydrazyl reagent (DPPH, free radical, 95%; Alfa Aesar, Karlsruhe, Germany). Samples (1 cm × 1 cm) of each film were placed in a 24-well plate and filled with 2 m of DPPH solution (250 µM solution in methyl alcohol), and left without exposure to light for 0.5 h. The absorbance of the samples (*Abs_PB_*) and the control (*Abs_DPPH_*) were measured spectrophotometrically at 517 nm (UV-1800, Shimadzu, Reinach, Switzerland). The radical scavenging assay was calculated from the formula:(1)$$the\;RSA\;(\%)=\frac{{Abs}_{DPPH}-{Abs}_{PB}}{{Abs}_{DPPH}}*100$$

#### 2.4.4. Roughness of Surface

The surface roughness of the films (1 cm × 1 cm) was analyzed at room temperature, using a microscope with a scanning SPM probe of the NanoScope MultiMode type (Veeco Metrology, Inc., Santa Barbara, CA, USA), which operated in a tapping mode. Two parameters—the root-mean-square roughness (Rq) and the arithmetic mean (Ra)—were measured (n = 5) using the Nanoscope v6.11 software (Bruker Optoc GmbH, Ettlingen, Germany).

#### 2.4.5. Surface Free Energy

In this experiment, the contact angles of glycerin or diiodomethane were measured at a constant temperature value using a goniometer equipped with a drop shape analysis system (DSA 10 Control Unit, Krüss, Hamburg, Germany). The surface free energy IFT(s), polar IFT(s,P), and dispersive IFT(s,D) components were calculated using the Owens–Wendt method.

#### 2.4.6. Water Vapor Permeation Rate (WVPR)

A dried anhydrous calcium chloride (m_0_), to be used as a desiccant, was placed in a plastic container (5 cm diameter). The films were placed onto the desiccant, and the container was sealed tightly. After three days, the calcium chloride was weighed (m_t_) and the change in its weight was determined, which was considered as water vapor absorbed by the desiccant. Then, WVPR was calculated in mg/cm^2^/h.

### 2.5. Statistical Analysis

Obtained results were expressed as the mean plus standard deviation (x ± SD), and were statistically analyzed using commercial software (SigmaPlot 15.0, Systat Software, San Jose, CA, USA). The normal distribution of data was checked using the Shapiro–Wilk test. One-way ANOVA analysis was performed, with multiple comparisons to the control using the Bonferroni *t*-test, with *p* < 0.05.

## 3. Results and Discussion

### 3.1. Graphene Nanoparticle Characterization

All material characterization was multidirectional. Figure 1A,B depict the SEM images of the GF15 and GF75 samples. As shown, the graphene flakes were clean, without loose particles on the surface. Figure 2 presents the SEM/STEM mode images. In particular, the image for the GF15 sample shows very thin graphene layers; so thin that the copper mesh can be seen. STEM also confirms the purity of the material, which is especially important for biocompatibility testing. The remaining images in Figure 2B–D show agglomerates of graphene flakes; they are also free of impurities.

Table 1 presents the content of three key elements, i.e., N, C, and H, and their porosity measurements (pore volume). C carbon C is the main component in all the samples under investigation. It ranges from 89.3 wt.% to 98.0 wt.%, which is typical for materials considered as pristine graphene. The rest of the content may be ascribed mainly to oxygen, the content of which is very low (far below the level typically occurring in the case of graphene oxide). The raw materials do not contain heavy metals, which has been proven in our previous works [20,21]. The results show that the materials are free of unnecessary impurities.

Nitrogen adsorption–desorption isotherms (Figure 3) for all samples belong to type II (IUPAC standard), which represents the unrestricted monolayer–multilayer adsorption process. It is probable that multilayer adsorption occurs in mesoporous materials, which contributes to the total pore volume. The surface area increased from 145 m^2^/g to 750 m^2^/g. Usually, an increase in surface area results from a diminishing of graphene plate size [21]. Also, the percentage of the mesopore volume V_me_ in the total pore volume V_t_ increased from 44 to 87%. Moreover, the sample with the highest surface area (GF75) does not have the best biocompatibility features.

The Raman spectra of the investigated samples (Figure 4A) show the typical shape for graphene peaks. G peak intensity corresponds to the degree of graphitization. The graphene nanoplatelets’ Raman spectra are characteristic because of two specific peaks at 1340 cm^−1^ (D band) and 1580 cm^−1^ (G band) [22]. However, despite all structural imperfections and irregularities, some similarly agglomerated graphene domains are present in all materials under investigation. It may be concluded that a few-layered graphene (FLG) is a dominating form, in which graphene flakes are self-organized.

In turn, in Figure 4B, the intensity ratios of the D to G bands (I_D_/I_G_) depend mainly on the level of the disorder. The I_D_/I_G_ ratio for GF15 is 0.23, GF30 is 0.45, GF50 is 0.52, and GF75 is 0.62. The interpretation of the I_D_/I_G_ ratios states that the amount of defects in the graphene nanoparticles is small.

### 3.2. In Vitro Biocompatibility

#### 3.2.1. Hemocompatibility

All tested nanoparticles damaged the integrity of human erythrocytes (Figure 5), causing a significant increase in hemolysis rate. A negative trend due to a smaller surface area can be found. Further, it was observed that all the analyzed groups significantly increased the percentage of hemolysis compared to the control condition. To sum up, nanoplatelets with 150 m^2^/g surface area had a hemotoxic effect, and medium (300 and 500 m^2^/g) were slightly hemolytic, while only the 75 GF were classified as nonhemolytic [17].

The effect of graphene and its derivatives on hemolysis has been previously tested and discussed. Research has shown that several factors may affect the hemocompatibility of particles, such as their size and shape; purity; concentration and dispersion; surface charge and functionalization; stability; and, finally, exposure time and environmental conditions [23,24]. Further, Sasidharan A. et al. [25] found that graphene nanomaterials with doses of up to 75 μg/mL did not elicit hemolysis, however, a negative trend with increasing particle concentration was observed. Here, the applied concentration was much higher (~33.3 mg/mL), and the particles’ shape was also different (here, we applied nanoplatelets), which may explain the differences in results. Further, it is assumed that various nanoparticles might affect the erythrocyte membrane integrity through mechanical damage or the generation of reactive oxygen species [26].

#### 3.2.2. Cytocompatibility

Nanoparticles did not negatively affect the viability of osteoblasts (tested through extract exposure for 24 h), which was confirmed by comparable MTT results (Figure 6). All groups showed high cytocompatibility (close to 100%), although a slightly significant increase of LDH release was noticed in the hFOB cell treated with extracts, especially for the nanoplatelets with the smallest surface area (GF15).

Carey et al. [27] also confirmed the cytocompatibility of the graphene flakes with human umbilical vein endothelial cells (up to 1 mg/mL), which is consistent with our results. Further, Chang et al. [28] reported no cytotoxic effect on lung carcinoma epithelial cells caused by graphene oxide (up to 0.2 mg/mL). Also, Guo et al. [29] developed GO-coating, which showed good compatibility with MC3T3-E1 osteoblasts and even promoted osteogenic differentiation. However, some reports have been made regarding the cytotoxic effects of graphene and its derivatives. For example, Wang et al. observed that a concentration of graphene oxide above 50 μg/mL had a cytotoxic effect on human fibroblast cells [30], and Ricci et al. [31] found that graphene nanoribbons were toxic above 200 µg/mL for MG-63. In conclusion, based on the literature [26], it can be assumed that the shape and size of particles and their concentration significantly impacts cytocompatibility properties. Further, the size-dependent toxicity between erythrocytes and human cells was previously noted in some reports regarding various nanoparticles [32,33]. Also, Liao et al. found that various graphene types showed different biotoxicity results, probably due to surface area and hydrophobic surfaces [34]. Moreover, differences between our hemo- and cytocompatibility results may also be related to the applied method. In the hemolysis study, the nanoparticles were in direct contact with cells, while in the cytocompatibility study, the extracts were used.

### 3.3. Film Characterization

Gelatin-based films were obtained through solvent evaporation and were modified with the addition of GF30, GF50, and GF75 (Figure 7). We decided not to use GF15, as this group showed the greatest hemolysis rate and an increase of released LHD in the cytocompatibility study.

#### 3.3.1. Scanning Electron Microscope (SEM)

Scanning electron microscope images of gelatin-based films with graphene nanoparticles are shown in Figure 8 and Figure 9. It is observed that graphene is totally embedded in the matrix and well distributed in the whole volume of gelatin. However, the morphology of the surface changes and is rougher than gelatin film without graphene.

#### 3.3.2. Mechanical Properties

Gelatin-based films containing 10% graphene nanoplatelets showed higher mechanical properties than unmodified films (Figure 10). The Young’s modulus for Gel_GF75 was twice the value for pure gelatin film, while the maximum tensile strength was triple. This suggests that the mechanical strength of the obtained films was significantly improved after the modification, are they are much stronger than they were beforehand. These properties are essential for appropriately applying wound dressing materials to the injury. If the film does not have adequate resistance to mechanical stresses occurring during handling, it will be unsuitable for medical use. Hence, the modification of nanoplatelets in this aspect is very beneficial. The positive effect of the addition of nanoparticles was previously noted in the literature. For example, Wang et al. reported that adding graphene oxide to gelatin increased the mechanical parameters of the obtained films [8].

#### 3.3.3. Antioxidant Activity

In the literature, attention is increasingly paid to nanoparticles in the context of a compound with a strong antioxidant effect. Graphene-modified materials have an antioxidant effect thanks to scavenging DPPH radicals, which can release free radicals and form non-radical species [35]. The antioxidant results of the obtained gelatin films with graphene nanoplatelets are presented in Table 2. In modified films with specific types of graphene, a significant increase in the RSA parameter was observed compared to the control unmodified gelatin film, which does not show any markers of antioxidant activity. It is worth noting that the antioxidant effect increases with a higher surface area of nanoplatelets, and the RSA parameter characteristic was the greatest in the Gel_GF75 film.

#### 3.3.4. Roughness of Surface

Both Ra and Rq increased after the addition of graphene nanoparticles (Table 3). The roughness of the films’ surface increases with increasing surface area of GF. The surface roughness can be classified as nanoroughness (less than 100 nm). The morphology of the obtained gelatin-based films is shown in Figure 11. To consider the material’s biomedical applicability, a rough surface should be characterized as a requirement, as it improves the cell’s adhesion to the film due to its flexible cell membrane. Moreover, the bacteria’s attachment to the film surface is also dependent on roughness [36]. Therefore, a positive effect of film modification on surface differentiation was found in this study.

#### 3.3.5. Surface Free Energy

The results presented in Table 4 show that increasing the surface area of graphene in the obtained films reduces both the surface free energy and the dispersion component, while increasing the polar component. Lowering the surface free energy parameter may result in improved cell–material interactions, which is essential from the point of view of using materials as dressings for wound treatment. In the literature, it can be found that the hydrophobicity of the graphene surface significantly affects the decrease in surface energy, and that at room temperature, the surface energy is about 46.7 mJ/m^2^ [37]; this is a value close to the surface energy value shown in Table 4.

#### 3.3.6. Water Vapor Permeation Rate (WVPR)

Analyzing the results in Table 5, it is noticeable that the water vapor permeability gradually decreased with the increase in the graphene content in the obtained materials. Gelatin is a material with a hydrophilic nature [38], which allows water molecules to bind and, as a result, allows water to penetrate through the created film. However, the addition of graphene causes a decrease in the WVPR parameter because graphene has hydrophobic properties [37]. Also, the decrease in the WVPR parameter may be related to the increased resistance to water penetration, because an increase in graphene concentration leads to an increase in the thickness of the membrane, which hinders water vapor penetration [39].

## 4. Conclusions

In this study, novel gelatin films were obtained by adding graphene nanoplatelets with different surface areas at 10% concentration. Before obtaining the films, four groups of GF with differing surface areas in the range of 150–750 m^2^/g were characterized. We found that they differ in morphology, porous structure, and average pore volume. Further, we checked their biological properties, including hemo- and cytocompatibility. Due to potential toxicity in medical applications, we selected only three groups of GF30, 50, and 75 as film modifications for precise evaluation. These additives did not affect the process of obtaining films, and we managed to produce interesting materials for wound healing use. The proposed films modified by graphene nanoplatelets were characterized by greater surface diversity; increased antioxidant activity; decreased surface free energy; and, finally, improved mechanical strength. However, we have unfortunately noticed a deterioration of the water vapor permeation rate, which is related to wound moisture and has an essential impact on tissue regeneration. Due to the performed research, we recommend the film modified with graphene platelets with a surface area of 500 m^2^/g–Gel_GF50 as the optimal candidate for wound dressing applications.

## Figures and Tables

**Figure 1 nanomaterials-13-03068-f001:**
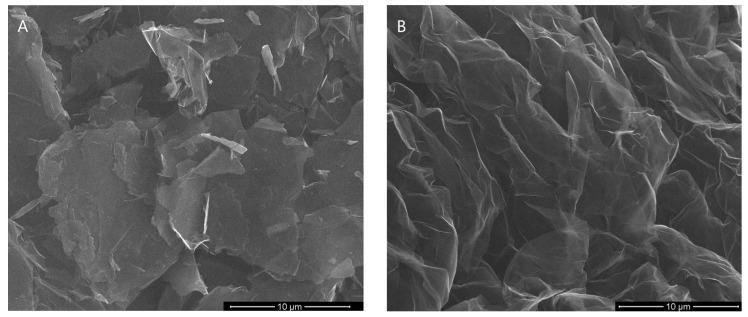
The morphology of GF15 (**A**) and GF75 (**B**) samples.

**Figure 2 nanomaterials-13-03068-f002:**
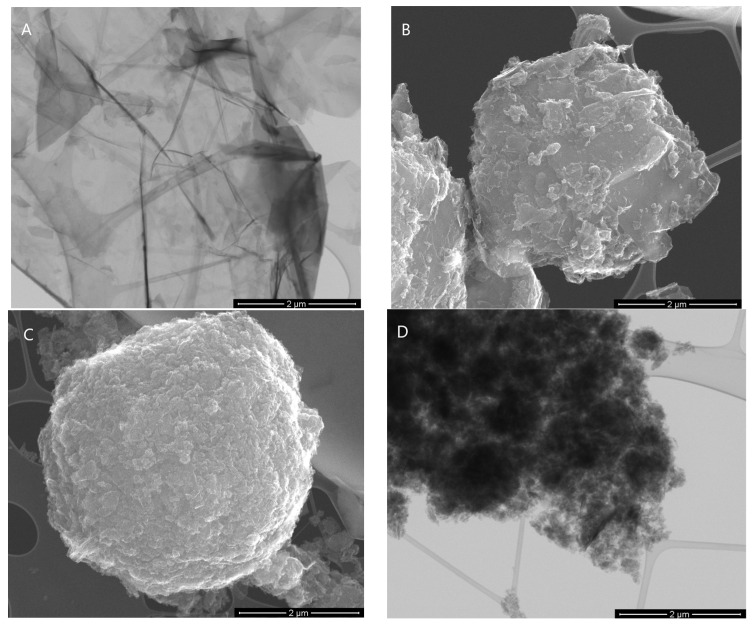
The morphology of GF15 (**A**), GF30 (**B**), GF50 (**C**), and GF75 (**D**) nanoparticles.

**Figure 3 nanomaterials-13-03068-f003:**
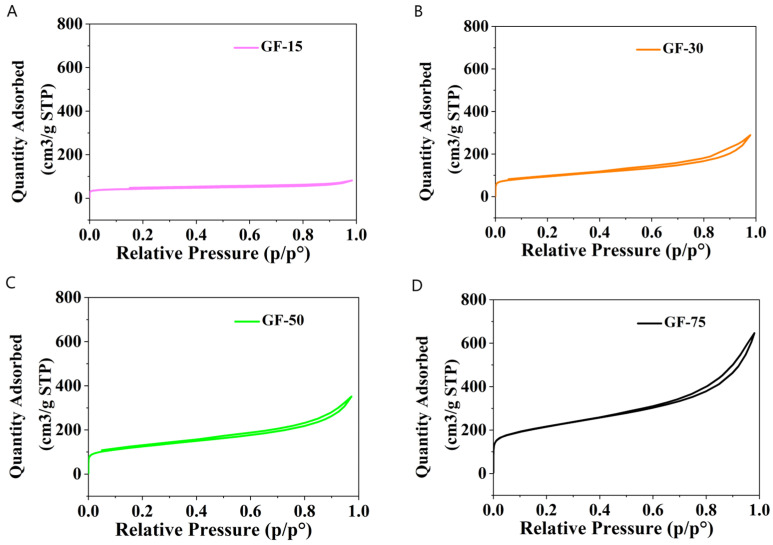
Adsorption–desorption isotherms of samples (**A**) GF15, (**B**) GF30, (**C**) GF50, and (**D**) GF75.

**Figure 4 nanomaterials-13-03068-f004:**
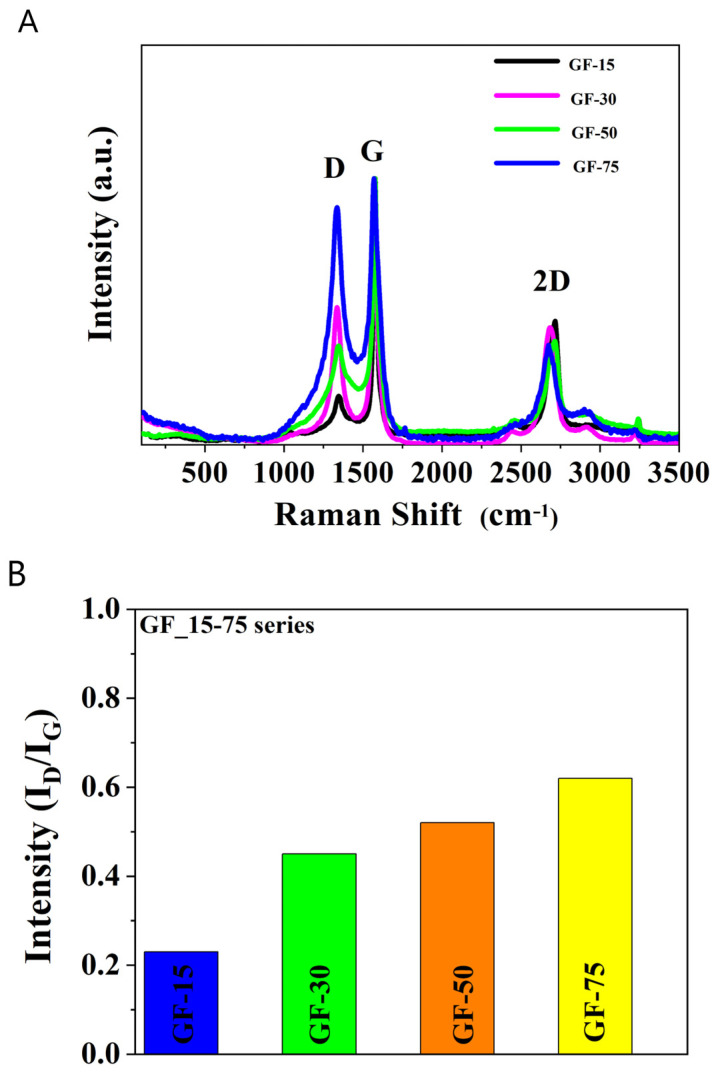
The intensities ratio of the D and G bands for GF15, GF30, GF50, and GF75. (**A**) Raman spectra of the D and G bands for GF15, GF30, GF50, and GF75; (**B**) The intensities ratio of the D and G bands for GF15, GF30, GF50, and GF75.

**Figure 5 nanomaterials-13-03068-f005:**
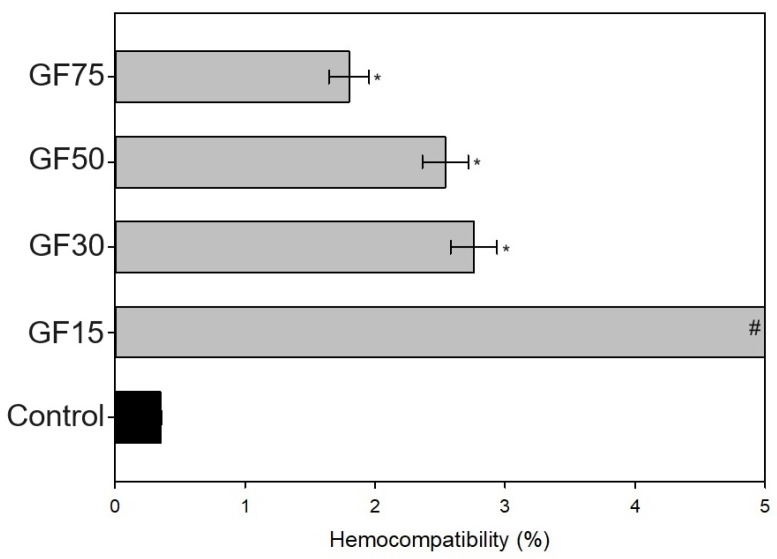
The effect of tested nanoparticles on the hemocompatibility of human erythrocytes (percentage hemolysis rate) after 24 h exposure (n = 3; data are expressed as the mean ± SD; * significantly different from the negative control (*p* < 0.05); # measurement outside the device’s range, above 5%).

**Figure 6 nanomaterials-13-03068-f006:**
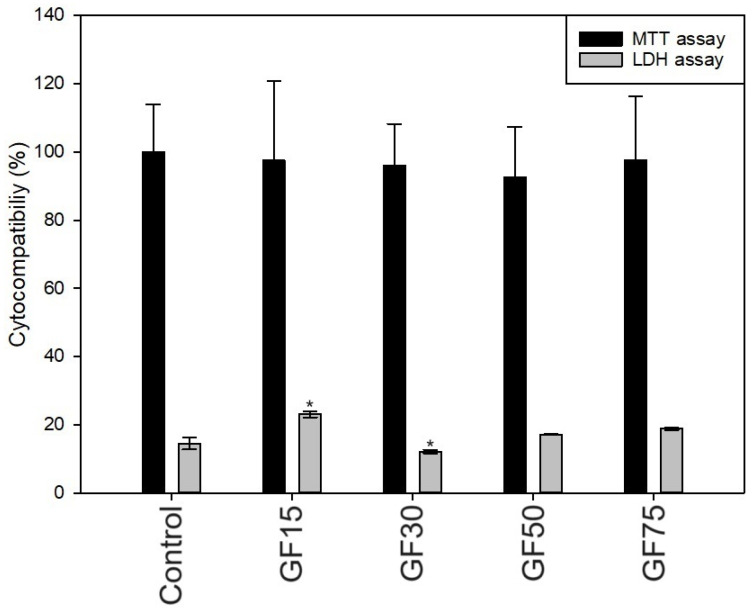
The effect of tested nanoparticles on the cytocompatibility of hFOB 1.19 cells (cell viability and lactate dehydrogenase release) after 24 h of exposure to sample extracts (n = 3; data are expressed as the mean ± SD; * statistical significance compared to the control–TCP-*p* < 0.05).

**Figure 7 nanomaterials-13-03068-f007:**
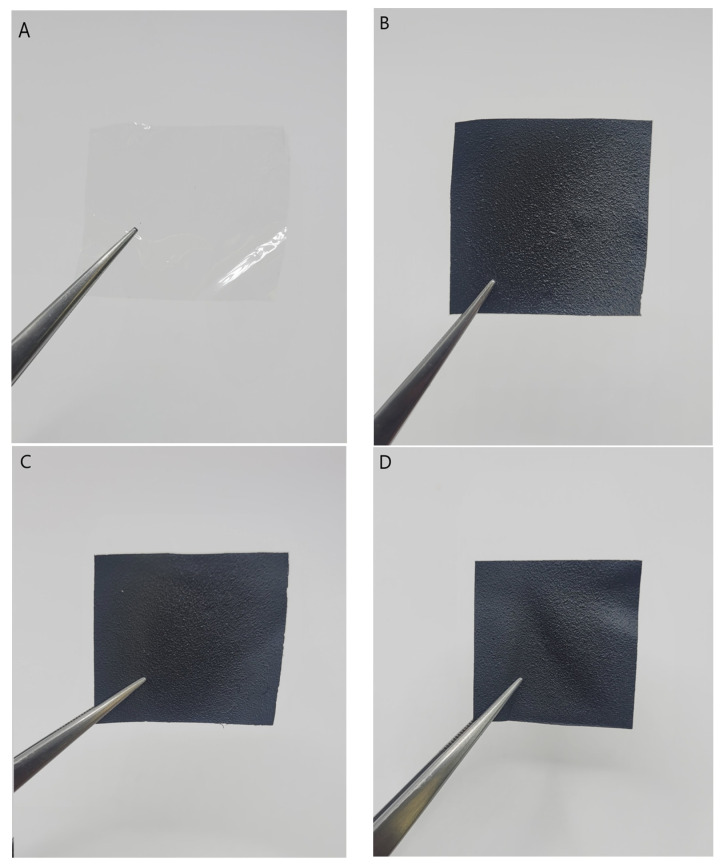
The images of the obtained films. (**A**) Gel; (**B**) Gel_GF30; (**C**) Gel_GF50; (**D**) Gel_GF75.

**Figure 8 nanomaterials-13-03068-f008:**
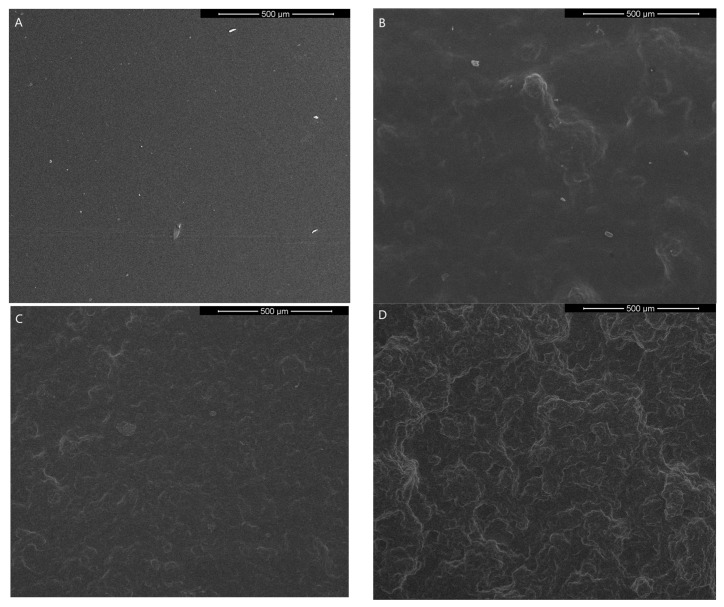
The SEM images of the surface morphology of the obtained films—(**A**) Gel, (**B**) Gel_GF30, (**C**) Gel_GH50, (**D**) Gel_GF75 (mag. 200×).

**Figure 9 nanomaterials-13-03068-f009:**
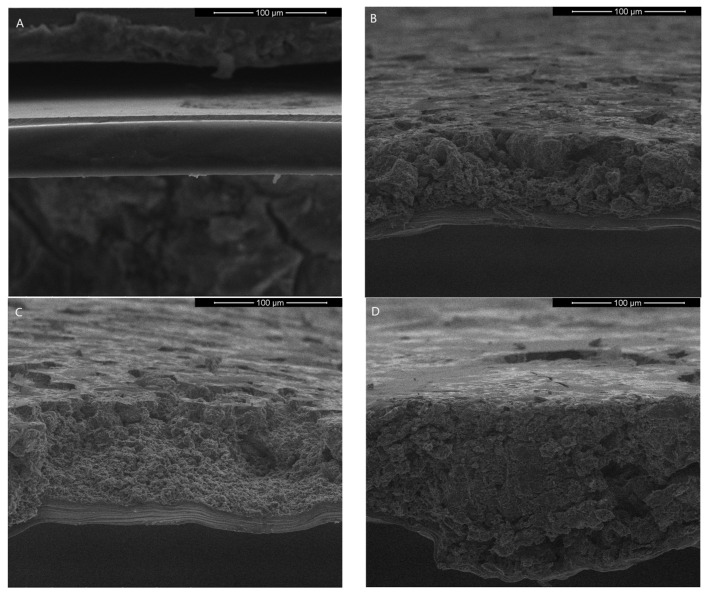
The SEM images of the cross-section morphology of the obtained films—(**A**) Gel, (**B**) Gel_GF30, (**C**) Gel_GH50, (**D**) Gel_GF75 (mag. 1000×).

**Figure 10 nanomaterials-13-03068-f010:**
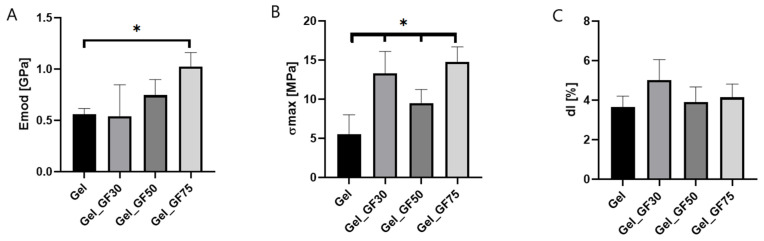
The mechanical properties of tested films: Young’s modulus (**A**), maximum tensile strength (**B**), and maximum elongation (**C**) (n = 10; * significantly different from control-Gel *p* < 0.05).

**Figure 11 nanomaterials-13-03068-f011:**
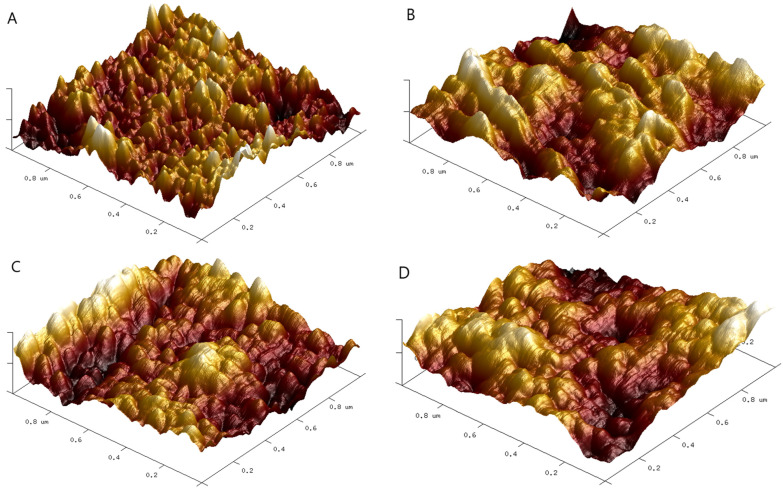
3D images of films’ surface of gelatin (**A**), gelatin with GF30 (**B**), gelatin with GF50 (**C**), and gelatin with GF75 (**D**).

**Table 1 nanomaterials-13-03068-t001:** The content of C, H, N, and surface parameters of the used carbons.

Sample	Elemental Content (wt%)				S_BET_	V_t_	V_mi_	V_me_	V_me_/V_t_
	N	C	H	Rest	(m^2^/g)	(cm^3^/g)	(cm^3^/g)	(cm^3^/g)	(%)
GF-15	0.6	90.8	0.9	7.7	145	0.256	0.178	0.078	44
GF-30	0.3	98.0	0.5	1.2	326	0.416	0.156	0.260	63
GF-50	0.5	91.7	0.6	7.2	431	0.678	0.139	0.539	79
GF-75	0.7	89.3	0.9	9.1	750	0.999	0.127	0.873	87

**Table 2 nanomaterials-13-03068-t002:** The radical scavenging assay (RSA) of gelatin films with graphene nanoparticles (n = 5; * significantly different from Gel—*p* < 0.05).

Specimen	RSA [%]
Gel	2.67 ± 0.02
Gel_GF30	35.70 ± 0.06 *
Gel_GF50	42.42 ± 0.11 *
Gel_GF75	46.14 ± 0.05 *

**Table 3 nanomaterials-13-03068-t003:** Roughness parameters (Ra and Rq) of gelatin films with graphene nanoparticles (n = 5; * significantly different from Gel—*p* < 0.05).

Specimen	Ra [nm]	Rq [nm]
Gel	1.75 ± 0.14	2.22 ± 0.21
Gel_GF30	1.83 ± 0.17	2.27 ± 0.13
Gel_GF50	2.22 ± 0.07 *	2.80 ± 0.03 *
Gel_GF75	2.47 ± 0.19 *	3.25 ± 0.16 *

**Table 4 nanomaterials-13-03068-t004:** The contact angle of water (Θ**^W^**, diiodomethane (Θ^D^), surface free energy (IFT(s)), and the polar (IFT(s,D)) and dispersive (IFT(s,D)) components of films, based on gelatin with and without graphene nanoparticles (n = 5; * significantly different from Gel—*p* < 0.05).

Specimen	Θ^W^ [^o^]	Θ^D^ [^o^]	IFT(s) [mJ/m^2^]	IFT(s,D) [mJ/m^2^]	IFT(s,P) [mJ/m^2^]
Gel	78 ± 1.12	21.15 ± 0.95	42.11 ± 0.11	35.40 ± 0.31	7.20 ± 0.19
Gel_GF30	64.43 ± 1.40 *	41.83 ± 0.23 *	40.51 ± 0.20 *	31.93 ± 0.08 *	8.58 ± 0.12 *
Gel_GF50	63.45 ± 0.60 *	47.33 ± 1.16 *	38.48 ± 0.38 *	28.75 ± 0.26 *	9.73 ± 0.12 *
Gel+GF75	54.67 ± 2.28 *	62.70 ± 1.30 *	36.88 ± 0.75 *	23.88 ± 0.47 *	13.00 ± 0.28 *

**Table 5 nanomaterials-13-03068-t005:** The water vapor permeability rate, recalculated to mg/cm^2^/h units (WVPR) (n = 5; * significantly different from Gel—*p* < 0.05).

Specimen	WVPR [mg/cm^2^/h]
Gel	0.114 ± 0.021
Gel_GF30	0.108 ± 0.026
Gel_GF50	0.102 ± 0.030
Gel+GF75	0.072 ± 0.01 *

## Data Availability

The data used to support the findings of this study can be made available by the corresponding author upon request.

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
