# Peer review of "Evaluating Gelatin-Based Films with Graphene Nanoparticles for Wound Healing Applications"

_nanomaterials, 2023, doi:10.3390/nano13233068_

Round 1
Reviewer 1 Report
Comments and Suggestions for Authors
The authors introduce several commercial graphene into a biofilm and then test it as wound healing.
In my opinion the authors should clearly state why graphene and why the percentage used? The authors also highlight the importance of the porosity structure in the introduction. Later on they do not relate the result with the porosity. It is recommendable to represent the main result of the biopolymer vs the porosity properties such us surface area, mico/mesopore volume, etc.
Others important properties of the material in this concrete application is the water affinity and surface groups (Infra red spectrums)
Some minor comments
From abstract It is understood that the biofilm is characterized in morphology by SEM and STEM. When it is not, only the commercial nanoparticles were characterized. I highly recommend characterizing the biofilm, the cross-section though SEM-EDX, and evaluate the thickness and composition.
2.4. Film preparation and characterization
Specify if 1% is in weight or volume
Why authors selected 10% instead on other?
Specify rpm of magnetic stirring
How much mixture is place in the holders? Please specify the weight or volumen
Line 141 says “Thin films with 0.017 mm (± 0.003) thickness were obtained” How the authors know the thickness of the film?
Figure 2. As less layer stuck on graphene larger is the surface area. It seems that author found the opposite, as GF15 is the thinnest materials. Please double check, because it goes against the theory.
Author Response
Dear All,
on behalf of myself and co-authors, I am enclosing the manuscript nanomaterials-2714030 entitled “Gelatin-based films with graphene nanoparticles for wound healing application” that we believe should be of strong interest to the general readership of the Nanomaterials.
We would like to note that in addition to addressing all reviewer’s valuable remarks, the authors placed additional editorial corrections including references to improve the quality of the manuscript. Below are our point-by-point responses to reviewer’s comments:
Reviewer #1:
- In my opinion the authors should clearly state why graphene and why the percentage used? The authors also highlight the importance of the porosity structure in the introduction. Later on they do not relate the result with the porosity. It is recommendable to represent the main result of the biopolymer vs the porosity properties such us surface area, mico/mesopore volume, etc.
Thank you very much for the comment. The percentage of graphene was proposed as 10% as in lower amount we did not observe the obtainment of homogeneous film. We have shown that porosity is important when selecting the appropriate raw carbon material. Unfortunately, the biopolymer film is very soft and cannot be turned into a powder for porosity measurements.
- Others important properties of the material in this concrete application is the water affinity and surface groups (Infra red spectrums)
Thank you for this comment. We added the contact angles of water and dijodomethane that show the water affinity of materials (table 4). It is not possible to do FTIR. We have tried by the sample is not transpartent (figure 7). For example, the obtained spectrum for Gel_GF50 looks like:

- From abstract It is understood that the biofilm is characterized in morphology by SEM and STEM. When it is not, only the commercial nanoparticles were characterized. I highly recommend characterizing the biofilm, the cross-section though SEM-EDX, and evaluate the thickness and composition.
Thank you very much for this valuable comment. We added the SEM images of obtained films (Figure 8 and 9). We hope it is acceptable now.
- 2.4. Film preparation and characterization
4.1. Specify if 1% is in weight or volume
Thank you. It is “1 w/w%”
4.2. Why authors selected 10% instead on other?
Thank you very much for the comment. With lower concentration of graphene we did not obtain homogeneous films that in our opinion would be difficult to characterize. It is now written in “Film preparation and characterization” section:
“Selected graphene nanoparticles were added to the gelatin solution at 10 w/w% con-centration that was the lowest concentration that allowed to obtain homogeneous films.”
4.3. Specify rpm of magnetic stirring
Thank you. It was 400 rpm.
4.4. How much mixture is place in the holders? Please specify the weight or volume
Thank you. It is now written “placed in plastic holders (40 mL per 10 cm x 10 cm)”
4.5. Line 141 says “Thin films with 0.017 mm (± 0.003) thickness were obtained” How the authors know the thickness of the film?
Thank you for this comment. Thickness was measured using thickness gauge. It is now written:
“Thin films with 0.017 mm (± 0.003) thickness measured with a gauge (Sylvac, Valbirse, Switzerland) were obtained.”
4.6. Figure 2. As less layer stuck on graphene larger is the surface area. It seems that author found the opposite, as GF15 is the thinnest materials. Please double check, because it goes against the theory.
Thank you very much for the comment. However, it is a subjective matter. According to our experience, there is not such a direct relation as suggested by Reviewer 1 because we always should keep in mind the phenomenon of GR grain stacking. Even when the agglomeration level of the GR layer in a single GR grain is low, but such grains are intensively stacked the surface area gets diminished. That may be random and does not proceed in a single direction. Thus, such a random stacking does not lead to the increase of agglomeration degree in single GR grains.

Reviewer 2 Report
Comments and Suggestions for Authors
The proposed manuscript presents a study of gelatin-based films containing graphene nanoparticles for wound healing. Below are some comments highlighting general as well as more specific issues. The authors should carefully consider these comments before proceeding with publication.
General comments:
There is a fairly strong discussion on the biocompatibility of gelatin, and in particular its non-toxicity, which has been recognised by the FDA. On the other hand, the discussion on graphene is not very advanced. The authors state that 3D-structured graphene flakes are biocompatible with blood cells. Here, only one study is cited (reference 7), coming from the same authors. The work cited relates in part to the study of the rate of haemolysis when blood is brought into contact with graphene. It does not include a cytotoxicity study. On the other hand, it is possible to find several scientific articles demonstrating the hazardous nature of graphene, in particular this recent review article entitled "Health and safety perspectives of graphene in wearables and hybrid materials" (https://doi-org.docelec.univ-lyon1.fr/10.1016/j.jmst.2023.01.011), in particular the section entitled "Health hazards via dermal exposure". The authors should take the cited studies into account in the introductory section of the submitted paper, particularly with regard to direct contact of graphene with the skin.
While the graphene powder seem to be quite well characterised, there is not morphological characterization of the thin films showing the quality of the graphene dispersion inside the gelatin matrix. Hence, we do not know if the graphene is totally embedded in the matrix or if it flushes the surface, and if the flakes are strongly agglomerated or not. The shaping process can have a huge impact on the final properties of the composite. The authors should provide more information on this part. This should help to make a better analysis of the results regarding the roughness of the surface and the water vapor permeation rate.
More specific comments:
- Table 1 : « The rest to 100% may be ascribed mainly to 192 oxygen»? It is well known that graphene can contain elements like Fe, Cu, Ni, Co, Mo (https://doi.org/10.1002/anie.201106917). The authors should give the proof that these elements are not present in the samples they used, or modify the sentence by mentioning the possible presence of such metal elements. These metals can present a danger for the patient’s health.
- Films of 17 +/- 3 µm were prepared. Due to the weak value of the thickness, the authors should detail the protocol to perform the mechanical tests on those films because these technical aspects can strongly impact the quality of the results.
Comments on the Quality of English LanguageThere are minor editing of English language required.
Author Response
Dear All,
on behalf of myself and co-authors, I am enclosing the manuscript nanomaterials-2714030 entitled “Gelatin-based films with graphene nanoparticles for wound healing application” that we believe should be of strong interest to the general readership of the Nanomaterials.
We would like to note that in addition to addressing all reviewer’s valuable remarks, the authors placed additional editorial corrections including references to improve the quality of the manuscript. Below are our point-by-point responses to reviewer’s comments:
Reviewer #2:
- There is a fairly strong discussion on the biocompatibility of gelatin, and in particular its non-toxicity, which has been recognised by the FDA. On the other hand, the discussion on graphene is not very advanced. The authors state that 3D-structured graphene flakes are biocompatible with blood cells. Here, only one study is cited (reference 7), coming from the same authors. The work cited relates in part to the study of the rate of haemolysis when blood is brought into contact with graphene. It does not include a cytotoxicity study. On the other hand, it is possible to find several scientific articles demonstrating the hazardous nature of graphene, in particular this recent review article entitled "Health and safety perspectives of graphene in wearables and hybrid materials" (https://doi-org.docelec.univ-lyon1.fr/10.1016/j.jmst.2023.01.011), in particular the section entitled "Health hazards via dermal exposure". The authors should take the cited studies into account in the introductory section of the submitted paper, particularly with regard to direct contact of graphene with the skin.
Thank you for this suggestion. According to it, we analyzed the results from the proposed review and expanded both, the Introduction and the Discussion, as follows:
“However, there are also some reports about their cytotoxicity, dependent mainly on size, concentration, and time exposure [15]. This mechanism is attributed to the gener-ation of reactive oxygen stress, which is causing DNA damage or disturbing cell sig-naling [16]. For example, Shvedova et al. [17] reported that carbon derivatives may result in skin irritation and diseases after cutaneous exposure. Further, it is generally accepted that graphene shows higher cytotoxicity than graphene oxide, related mainly to its ag-gregation tendency [18].”
“Further, the size-dependent toxicity between erythrocytes and human cells was pre-viously noted in some reports regarding various nanoparticles [35,36]. Also, Liao et al. found that graphene types showed different results of biotoxicity, probably due to surface area and hydrophobic surface [37]. FurtherMoreover, differences between our the results between hemo- and cytocompatibility results testing may also be related to the applied method. In the hemolysis study, the nanoparticles were in direct contact with cells, while in the cytocompatibility study, the extracts were used.”
- While the graphene powder seem to be quite well characterised, there is not morphological characterization of the thin films showing the quality of the graphene dispersion inside the gelatin matrix. Hence, we do not know if the graphene is totally embedded in the matrix or if it flushes the surface, and if the flakes are strongly agglomerated or not. The shaping process can have a huge impact on the final properties of the composite. The authors should provide more information on this part. This should help to make a better analysis of the results regarding the roughness of the surface and the water vapor permeation rate.
Thank you very much for the comment. We have done additional morphological studies regarding the thin fils used in our research. We have added SEM images of films’ surface and the cross-section.
- -Table 1 : « The rest to 100% may be ascribed mainly to 192 oxygen»? It is well known that graphene can contain elements like Fe, Cu, Ni, Co, Mo (https://doi.org/10.1002/anie.201106917). The authors should give the proof that these elements are not present in the samples they used, or modify the sentence by mentioning the possible presence of such metal elements. These metals can present a danger for the patient’s health.
Thank you very much for the comment. It is true that commercial material may contain some minor amounts of elements different than carbon. In real samples of graphene-like materials carbon content exceeds 90% and still such materials are considered as “virgin” or “pristine” graphene. The second element for the intensity of occurrence is oxygen not to be detected in combustion elemental analysis which delivers only contents of C, N, H. In such a case, we attribute the residue to O exclusively since the content of heavy metals is very marginal. To find the presence of heavy metals we have performed in our previous works XPS analysis [22] and [23]. The raw materials do not have the heavy metals. Thus, the missing elemental content to complete 100% is ascribed to oxygen (….%).
4.- Films of 17 +/- 3 µm were prepared. Due to the weak value of the thickness, the authors should detail the protocol to perform the mechanical tests on those films because these technical aspects can strongly impact the quality of the results.
Thank you for this comment. We used protocol that was published before with other thin films also obtained by solvent evaporation. It is now added as reference and also method is now described in more detail.

Round 2
Reviewer 2 Report
Comments and Suggestions for Authors
Thank you to the authors for haviing taken into account the comments performed before.
Comments on the Quality of English LanguageSome parts of the document could be improved.